# Genetics of H5N1 and H5N8 High-Pathogenicity Avian Influenza Viruses Isolated in Japan in Winter 2021–2022

**DOI:** 10.3390/v16030358

**Published:** 2024-02-26

**Authors:** Junki Mine, Yoshihiro Takadate, Asuka Kumagai, Saki Sakuma, Ryota Tsunekuni, Kohtaro Miyazawa, Yuko Uchida

**Affiliations:** National Institute of Animal Health, National Agriculture and Food Research Organization, 3-1-5 Kannondai, Tsukuba 305-0856, Ibaraki, Japankumagaia412@affrc.go.jp (A.K.); sakumas438@affrc.go.jp (S.S.); tune@affrc.go.jp (R.T.); miyazawak@affrc.go.jp (K.M.); uchiyu@affrc.go.jp (Y.U.)

**Keywords:** avian flu, high pathogenicity avian influenza virus, Japan, 2021/2022 season, phylogenetic analysis

## Abstract

In winter 2021–2022, H5N1 and H5N8 high-pathogenicity avian influenza (HPAI) viruses (HPAIVs) caused serious outbreaks in Japan: 25 outbreaks of HPAI at poultry farms and 107 cases in wild birds or in the environment. Phylogenetic analyses divided H5 HPAIVs isolated in Japan in the winter of 2021–2022 into three groups—G2a, G2b, and G2d—which were disseminated at different locations and times. Full-genome sequencing analyses of these HPAIVs revealed a strong relationship of multiple genes between Japan and Siberia, suggesting that they arose from reassortment events with avian influenza viruses (AIVs) in Siberia. The results emphasize the complex of dissemination and reassortment events with the movement of migratory birds, and the importance of continual monitoring of AIVs in Japan and Siberia for early alerts to the intrusion of HPAIVs.

## 1. Introduction

The worldwide spread of H5 high-pathogenicity avian influenza (HPAI) viruses (HPAIVs) has caused huge outbreaks and economic losses in poultry over the past few years (World Organization for Animal Health: WOAH). Almost all recent H5 HPAIVs arose from A/goose/Guangdong/1/1996_(H5N1) (Gs/Gd lineage), which was first detected in China in 1996 [1,2] and evolved into multiple clades and subclades defined by hemagglutinin (HA) genes [3]. H5 HPAIVs in clade 2.3.4.4 emerged in 2013–2014 and became dominant among outbreaks worldwide [4]. In the evolution of this clade, N1 genes of the 2.3.4.4 H5 HPAIVs were replaced by various neuraminidase (NA) genes, such as N2, N3, N4, N5, N6, N8, and N9 [5].

In Japan, the first outbreaks of influenza caused by Gs/Gd-lineage clade 2.5 H5N1 HPAIVs occurred in 2004 [6], and multiple introductions of various clades of H5Nx HPAIVs were reported [7,8,9,10,11,12,13,14,15]. Clade 2.3.4.4b H5Nx HPAIVs were first reported in Japan in 2017 [16]. Japan experienced outbreaks caused by clade 2.3.4.4b H5N6 HPAIVs in winter 2017–2018, simultaneous with those caused by the same subtypes in Europe, followed by the intrusion of clade 2.3.4.4b H5N8 HPAIVs in the winter of 2020–2021 [17], which were further divided into subclusters G1 (those that circulated in Europe in early 2020) and G2 (those that circulated in Europe in late 2020) [18].

In 2021–2022 (July 2021–June 2022), numerous HPAI outbreaks caused by clade 2.3.4.4b H5 HPAIVs occurred in Japan. In total, 25 cases in poultry were reported; this was the third largest outbreak reported in Japan (following 84 cases in winter 2022–2023 and 52 in winter 2020–2021), and 1,890,000 birds were slaughtered to stamp it out (https://www.maff.go.jp/j/syouan/douei/tori/r3_hpai_kokunai.html (accessed on 15 January 2024)). Phylogenetic analyses using the HA gene of representative Japanese isolates in 2021–2022 revealed that the viruses were descendants of H5 HPAIVs in the G2 cluster in the winter of 2020–2021 and classified them into groups G2a, G2b, and G2d [19,20,21,22].

Here, we phylogenetically compared the full genomes of H5 HPAIVs that caused all outbreaks and some of the cases in wild birds in Japan in 2021–2022. We inferred the background of the dissemination of H5 HPAIVs in Japan, considering both the origins of all genes of the Japanese H5 HPAIVs in the winter of 2021–2022 and the movement of migratory birds.

## 2. Materials and Methods

### 2.1. Virus Isolation and Whole-Genome Sequencing

Tracheal or cloacal swabs collected at Japanese poultry farms with suspected HPAIV infections were inoculated into embryonated chicken eggs for virus isolation at the diagnostic laboratories of the livestock hygiene service centers in each farm’s prefecture. Allantoic fluid inoculated with swab samples that showed HA activity against chicken red blood cells and swab samples from dead wild birds in Hokkaido Prefecture were submitted to the National Institute of Animal Health (Tsukuba, Japan) for diagnosis.

The whole genomes of the isolated viruses were obtained [17]. In brief, cDNA libraries for next-generation sequencing were prepared with a NEBNext Ultra II RNA Library Prep Kit for Illumina (New England Biolabs, Ipswich, MA, USA) and sequenced with a MiSeq Reagent Kit v. 2 (Illumina, San Diego, CA, USA). Consensus sequences were generated in CLC Genomics Workbench (v. 9.5.3, Qiagen, Hilden, Germany) and FluGAS software (v. 2.2.5, World Fusion, Tokyo, Japan) [23]. The newly determined viral sequences have been deposited in the GISAID database (http://platform.gisaid.org (accessed on 15 January 2024)) with the accession numbers listed in Table 1.

### 2.2. Phylogenetic Analysis

In Japan in the winter of 2021–2022, a total of 132 HPAI reports were made, comprising 25 outbreaks at poultry farms and 107 positive identifications in wild birds or in the environment. We used strains isolated from all 25 outbreaks at poultry farms and 32 detections in wild birds in Hokkaido, whose diagnosis is responsible in the National Institute of Animal Health, for phylogenetic analysis. We downloaded the sequences of the avian influenza viruses (AIVs) from the NCBI (Influenza Virus Resource; https://www.ncbi.nlm.nih.gov/genomes/FLU/Database/nph-select.cgi?go=database) and GISAID databases (accessed on 13 June 2022). The GISAID sequences were aligned with those of AIVs stored at the National Institute of Animal Health in Japan in BioEdit v. 7.2.5 [24] and MAFFT v. 7.490 [25] software. Removal of sequences with ambiguous nucleotide bases left 100,361 sequences for genes for polymerase basic protein 2 (PB2), 98,773 for PB1, 103,136 for polymerase acidic protein (PA), 11,028 for H5, 105,489 for nucleoprotein (NP), 60,075 for N1, 5549 for N8, 108,555 for matrix protein (MP), and 109,352 for nonstructural protein (NS) for the phylogenetic analyses. Maximum-likelihood trees based on the aligned sequences were constructed in FastTree v. 2.1.10 software [26].

## 3. Results

### 3.1. Phylogenetic Analysis of Japanese H5 Isolates from the 2021–2022 Winter

Maximum likelihood analysis using the HA gene of the Japanese H5 HPAIVs isolated in the winter of 2021–2022 placed the viruses in three clusters in clade 2.3.4.4b—G2a, G2b, and G2d (Figure 1a)—which were previously named 20A, 20E, and 21E, respectively [21]. The average nucleotide sequence identity of the HPAIVs within a clade was >99.7%. That between the HPAIVs in G2a and G2b was 98.6%, and that between both of those and G2d was 97.9%.

G2a H5N8 HPAIVs isolated in November 2021 shared a common ancestor with viruses that were isolated in Japan, Korea, and China in the 2020–2021 winter (December 2020–February 2021) (Figure 1b). While almost all strains in this cluster were identified as H5N8, H5N6 isolates were also detected in humans in China during July to September 2021, implying the possibility of transmission from birds. G2a H5N8 HPAIVs were not detected in the nine months following February 2020 and might thus have been retained in the Asian continent and reintroduced into Japan in the 2021–2022 winter.

G2b H5N1 HPAIVs caused outbreaks from November 2021 to May 2022 (Figure 1c). The H5 HA genes shared a common ancestor with European H5N8 HPAIVs in the 2020–2021 winter, and those of Japanese isolates were closely related to the H5N8 HPAIVs isolated in the Novosibirsk region of Russia in August 2020 and May 2021.

G2d H5N1 HPAIVs caused outbreaks from February to May 2022 (Figure 1d). The H5 HA genes shared a common ancestor with European H5N1 HPAIVs in the 2021–2022 winter. The H5 HA genes of Japanese and European G2d H5N1 HPAIVs branched from those of H5N1 HPAIVs isolated in the Chelyabinsk, Novosibirsk, and Omsk regions of Russia during August and September 2021 as well as an isolate from Bangladesh in December 2021.

### 3.2. Characteristics of the Outbreaks in Japan during 2021–2022

In Japan, 25 outbreaks due to H5N1 and H5N8 HPAIVs were reported across 12 prefectures between 10 November 2021 and 13 May 2022 (Table 1; Figure 2a). Around the same time, 107 positive identifications in wild birds or in the environment were reported across seven prefectures (Appendix A). The three clusters of H5 HPAIVs isolated during the 2021–2022 winter had different detection periods. G2a H5N8 HPAIVs caused only two outbreaks, in Akita and Kagoshima, on 10 and 15 November, respectively, and none have been detected in Japan since (Figure 2b). G2b H5N1 HPAIVs were first detected in Kagoshima on 13 November and caused outbreaks throughout Japan (Table 1; Figure 2c). With the exception of the one in Hokkaido, outbreaks caused by G2b H5N1 HPAIVs were reported from November 2021 to January 2022. It is worth noting that the last case was reported 3½ months after the others (Table 1). Outbreaks caused by G2d H5N1 HPAIVs were reported in northern Honshu (Iwate, Miyagi, Aomori, and Akita) and Hokkaido during February to May 2022 (Figure 2d). Out of nine cases caused by G2d H5N1 HPAIVs, six were found in chickens and three in emus.

### 3.3. Gene Constellations of H5 HPAIVs Isolated in Japan in the 2021–2022 Winter

Phylogenetic analyses based on all RNA segments of the Japanese H5 HPAIVs revealed that almost all HPAIVs had internal genes with the same relationship as the HA genes: internal genes of the Japanese H5N8 HPAIVs formed a cluster with those of Asian and Russian isolates in 2020–2021, those of G2b H5N1 HPAIVs formed a cluster with the H5N8 HPAIVs in the 2020–2021 winter, and those of G2d H5N1 HPAIVs formed a cluster with the isolates that circulated in Europe in the 2021–2022 winter (Appendix A). Exceptionally, G2b H5N1 HPAIVs that caused outbreaks in Ehime during late December 2021 to early January 2022 had PB1 genes that were phylogenetically distant from other Japanese G2b H5N1 HPAIVs (Figure 3 and Figure 4) but close to those of AIVs isolated in Siberia. In addition, Japanese H5 HPAIVs had NA and internal genes that shared a common ancestor with some non-H5 AIVs. N1 genes of Japanese G2b H5N1 HPAIVs were closely related to that of the H1N1 AIV isolated in Novosibirsk in August 2020 (Figure 5), and AIVs in Siberia had internal genes that were phylogenetically related to those of Japanese G2b H5N1 HPAIVs (Figure 6).

## 4. Discussion

In the winter of 2021–2022, three phylogenetically distinct types of H5 HPAIVs (G2a, G2b, and G2d) intruded into Japan and caused serious outbreaks. Since the first case of HPAI in Japan in 2004 [6], that of 2021–2022 was the longest (November 2021–May 2022). The migration of multiple groups of wild birds might have contributed to the extended detection of H5 HPAIVs in 2021–2022. At the same time, at least two intrusions of H5 HPAIVs contributed to the extended outbreak: G2a and G2b H5 HPAIVs intruded around the start of November, and outbreaks caused by both were reported in northern and southern Japan. G2d H5 HPAIVs started to be detected in February 2022, and they caused outbreaks in limited areas in northern Japan. Along the East-Asia–Australian flyway lap in Japan [27], wild birds are considered to migrate to Japan along four routes [28]: the Kamchatka Peninsula—Kuril Islands, Sakhalin, Sea of Japan, and Korean Peninsula. Thus, the dissemination of G2d H5 HPAIVs might correspond to the flyways through the Kamchatka Peninsula—Kuril Islands and Sakhalin, resulting in their limited distribution. G2b H5 HPAIV caused an outbreak in Hokkaido in May 2022 after no detection for 3½ months. Spring movement northward via Hokkaido to Siberia might also contribute to the dissemination of viruses, resulting in the outbreak in spring.

Detection periods differed among clades: G2a H5 HPAIVs were detected only in November, but the other two were detected over 3 months. These differences might be due to the number of wild birds that are immunologically susceptible to G2a H5 HPAIVs. Phylogenetic analyses revealed that G2a H5 HPAIVs caused outbreaks in Asia in 2020–2021 [17,29,30,31] and were reintroduced in 2021–2022, while G2b and G2d H5 HPAIVs were detected for the first time in 2021–2022. These results imply that the migratory birds coming into Japan may have become more resistant to G2a H5 HPAIVs through previous exposure, resulting in the prevalent circulation of G2b and G2d H5 HPAIVs in 2021–2022. Further studies on antibodies and the higher prevalence of individual H5 HPAIVs among wild birds are needed to assess this explanation.

Several genes of H5 HPAIVs detected in Japan in 2021–2022 were phylogenetically related to those of the non-H5 AIVs in wild birds, indicating that they emerged through reassortment with AIVs. Notably, AIVs detected in Siberia had genes that were phylogenetically closely related to those of Japanese isolates (Figure 4 and Figure 5). These results suggest that the reassortment events happen in migratory birds in Siberia, the summer breeding range, and that the H5 HPAIVs are disseminated through southward migration, as reported previously [32,33].

G2d H5 HPAIVs caused outbreaks on poultry farms in northern Japan, including three emu farms. Full-genome analyses revealed that the G2d H5 HPAIVs isolated from emus had lysine (K) at position 627 in the PB2 segment while isolates from chickens had glutamic acid (E) at that position in the present study. The substitution E627K enhances replication and the pathogenicity of H5 HPAIVs in mammalian hosts [34,35], and the amino acid at position 627 in PB2 is reportedly exclusively glutamic acid in bird isolates and lysine in human isolates [36,37]. Internal tissues and organs of emus co-express α-2,3 and α-2,6 sialic acid receptors [38]. In the present study, emus are sensitive to avian-origin H5 HPAIVs and selected viruses with the substitution E627K in PB2, which can enhance replication in mammalian hosts, whereas our emu isolates had no substitutions that are deduced to increase the preference for α-2,6 sialic acid receptors. These results suggest that emus contribute to the emergence of new viruses with pandemic potential in humans. Further experiments to test host range and replication activity in cells are needed.

In summary, phylogenetic analyses revealed that H5 HPAIVs of three H5 HA genotypes caused influenza outbreaks in Japan during the 2021–2022 winter, each with a different mode of dissemination. Full-genome sequencing analyses revealed a strong relationship between viruses from Japan and Siberia, suggesting that the HPAIVs arose from reassortment events with AIVs in Siberia. Our results emphasize the complex of dissemination and reassortment events of HPAIVs with the movement of migratory birds and the importance of continual monitoring of AIVs in Japan and Siberia for early alerts to the intrusion of HPAIVs.

## Figures and Tables

**Figure 1 viruses-16-00358-f001:**
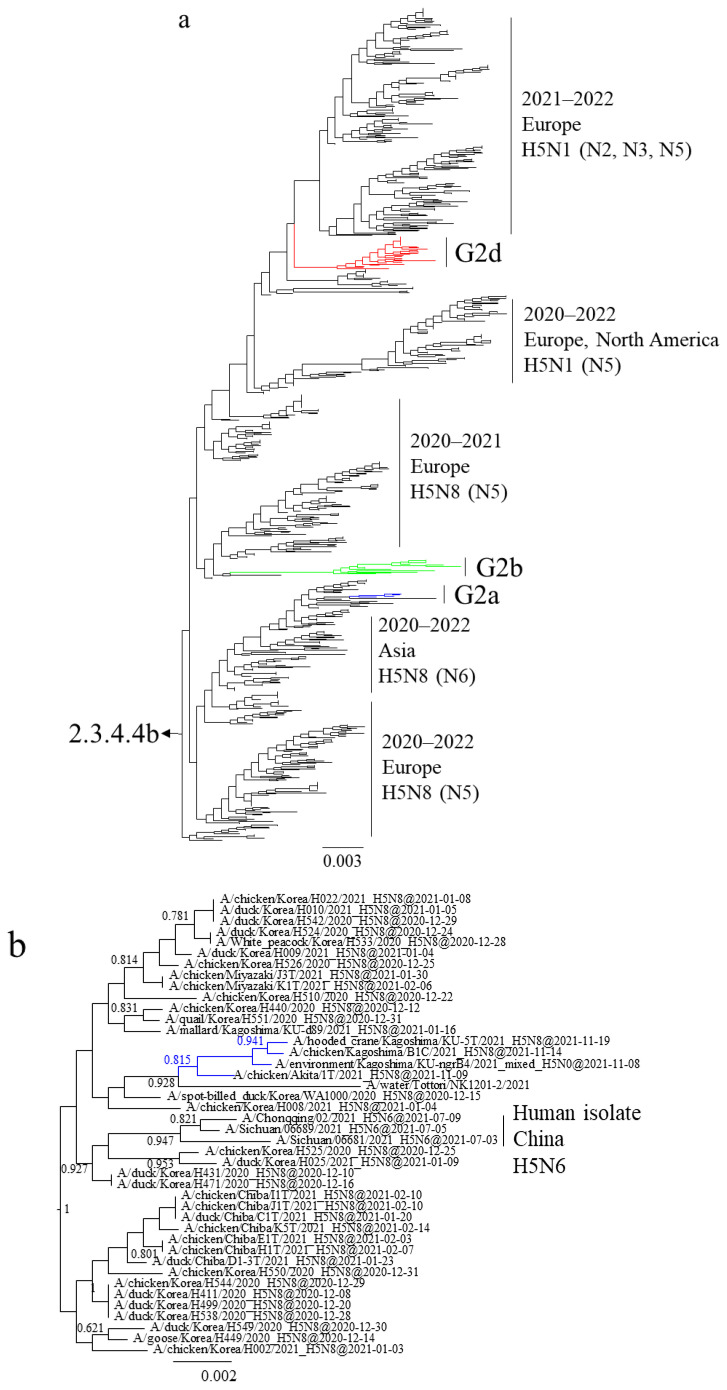
Section of the maximum-likelihood tree based on H5 HA genes. Colors indicate groups: blue, G2a; green, G2b; red, G2d. Sections of clade 2.3.4.4b covering (**a**) G2a, G2b, and G2d, (**b**) G2a, (**c**) G2b, and (**d**) G2d are shown. Fast-global bootstrap values of ≥0.60 are shown.

**Figure 2 viruses-16-00358-f002:**
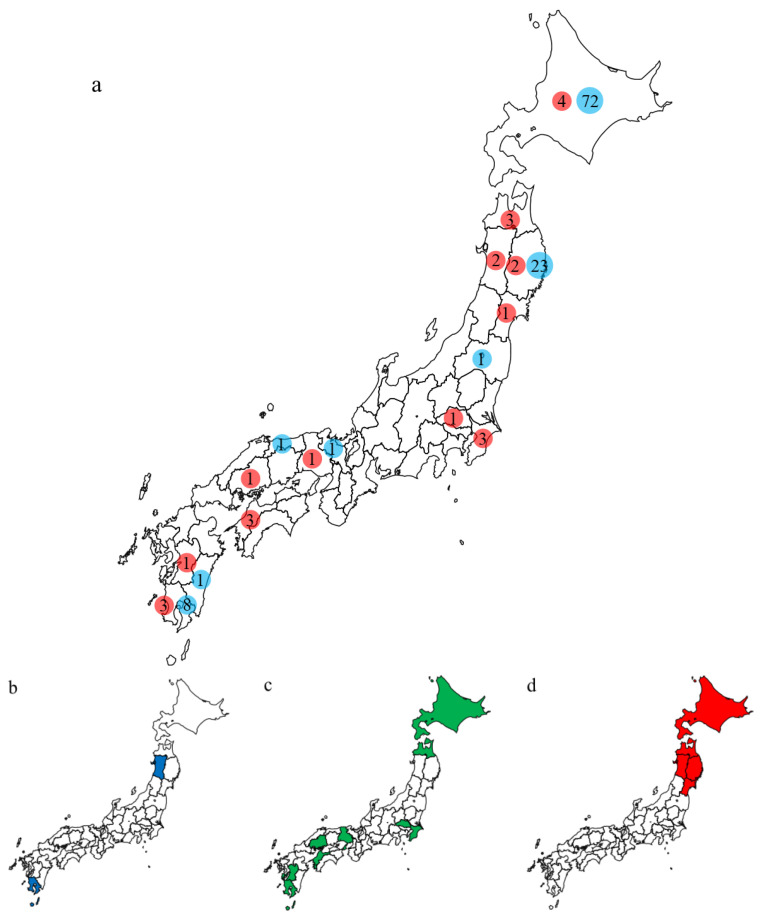
Geographic locations of prefectures where H5 HPAIVs were detected in Japan in the 2020–2021 winter. (**a**) Red, number of outbreaks; blue, number of detections in wild birds or in the environment. (**b**–**d**) Outbreaks caused by (**b**) G2a (blue), (**c**) G2b (green), and (**d**) G2d (red) H5 HPAIVs.

**Figure 3 viruses-16-00358-f003:**
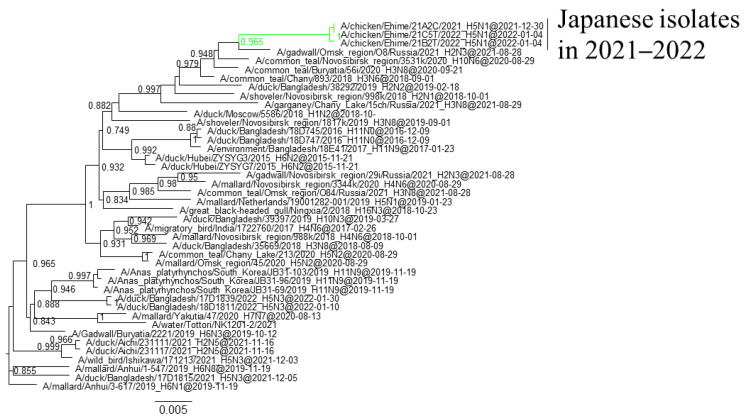
Section of the maximum-likelihood tree based on PB1 genes. G2b H5 HPAIVs isolated in Ehime Prefecture are in green. Fast-global bootstrap values of ≥0.60 are shown.

**Figure 4 viruses-16-00358-f004:**
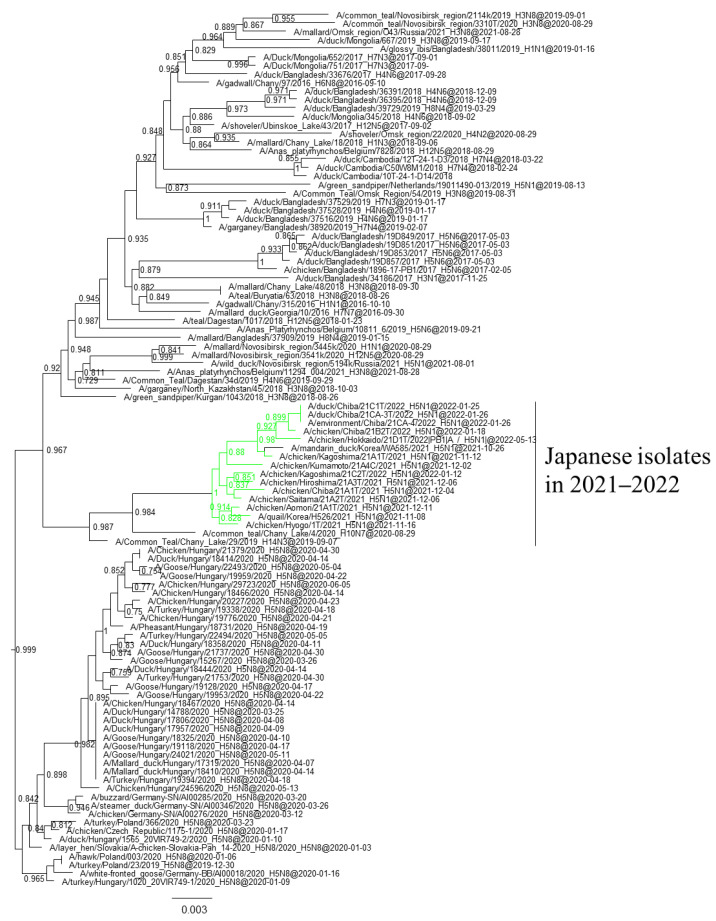
Section of the maximum-likelihood tree based on PB1 genes. G2b H5 HPAIVs isolated in this study are green. Fast-global bootstrap values of ≥0.60 are shown.

**Figure 5 viruses-16-00358-f005:**
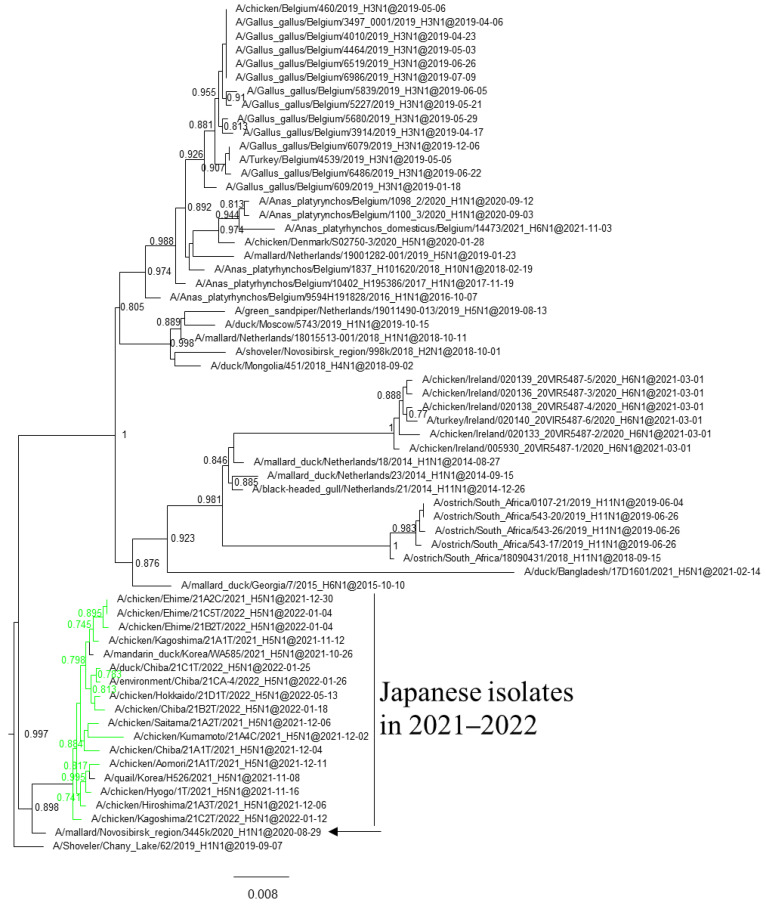
Section of the maximum-likelihood tree based on NA genes. G2b H5 HPAIVs isolated in this study are in green. Fast-global bootstrap values of ≥0.60 are shown. The arrow indicates the isolate in Siberia.

**Figure 6 viruses-16-00358-f006:**
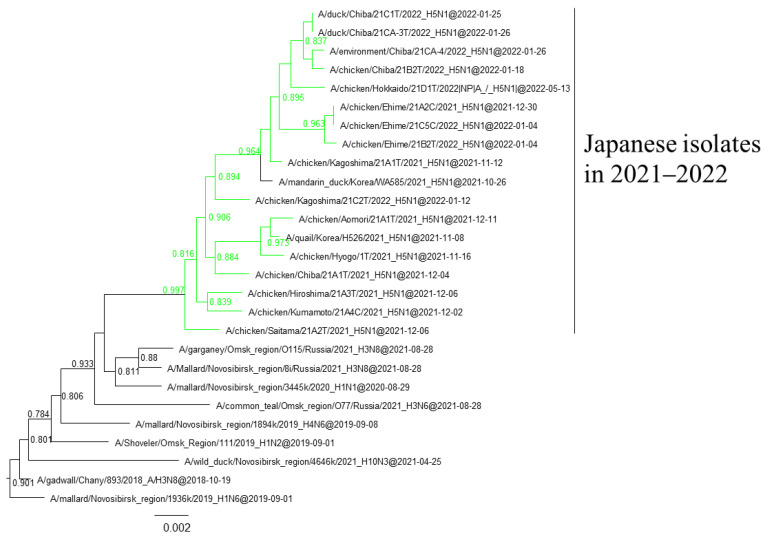
Section of the maximum-likelihood tree based on NP genes. G2b H5 HPAIVs isolated in this study are in green. Fast-global bootstrap values of ≥0.60 are shown.

**Table 1 viruses-16-00358-t001:** Information on H5 HPAIVs isolated from poultries in this study.

Genotype	Collection Date	Host	Prefecture	Area	Subtype	Representative isolate	Accession No.
G2a	10/11/2021	chicken	Akita	Northern	H5N8	A/chicken/Akita/1T/2021	EPI18720917
G2a	15/11/2021	chicken	Kagoshima	Kyushu	H5N8	A/chicken/Kagoshima/B1C/2021	EPI18720919
G2b	13/11/2021	chicken	Kagoshima	Kyushu	H5N1	A/chicken/Kagoshima/21A1T/2021	EPI18720695
G2b	17/11/2021	chicken	Hyogo	Western	H5N1	A/chicken/Hyogo/1T/2021	EPI18720691
G2b	3/12/2021	chicken	Kumamoto	Kyushu	H5N1	A/chicken/Kumamoto/21A4C/2021	EPI18720727
G2b	5/12/2021	chicken	Chiba	Central	H5N1	A/chicken/Chiba/21A1T/2021	EPI18720718
G2b	7/12/2021	chicken	Saitama	Central	H5N1	A/chicken/Saitama/21A2T/2021	EPI18720733
G2b	7/12/2021	chicken	Hiroshima	Western	H5N1	A/chicken/Hiroshima/21A3T/2021	EPI18720753
G2b	12/12/2021	chicken	Aomori	Northern	H5N1	A/chicken/Aomori/21A1T/2021	EPI18720743
G2b	31/12/2021	chicken	Ehime	Shikoku	H5N1	A/chicken/Ehime/21A2C/2021	EPI18720767
G2b	4/1/2022	chicken	Ehime	Shikoku	H5N1	A/chicken/Ehime/21B2T/2022	EPI18720769
G2b	4/1/2022	chicken	Ehime	Shikoku	H5N1	A/chicken/Ehime/21C5T/2022	EPI18720778
G2b	13/1/2022	chicken	Kagoshima	Kyushu	H5N1	A/chicken/Kagoshima/21C2T/2022	EPI18720780
G2b	19/1/2022	chicken	Chiba	Central	H5N1	A/chicken/Chiba/21B2T/2022	EPI18720761
G2b	26/1/2022	duck	Chiba	Central	H5N1	A/duck/Chiba/21C1T/2022	EPI18720794
G2b	13/5/2022	chicken	Hokkaido	Hokkaido	H5N1	A/chicken/Hokkaido/21D1T/2022	EPI18720843
G2d	11/2/2022	chicken	Iwate	Northern	H5N1	A/chicken/Iwate/21A1T/2022	EPI18720791
G2d	24/3/2022	chicken	Miyagi	Northern	H5N1	A/chicken/Miyagi/21A11T/2022	EPI18720811
G2d	7/4/2022	chicken	Aomori	Northern	H5N1	A/chicken/Aomori/21B3T/2022	EPI18720807
G2d	14/4/2022	chicken	Aomori	Northern	H5N1	A/chicken/Aomori/21C14T/2022	EPI18720816
G2d	15/4/2022	chicken	Hokkaido	Hokkaido	H5N1	A/chicken/Hokkaido/21A4T/2022	EPI18720826
G2d	16/4/2022	emu	Hokkaido	Hokkaido	H5N1	A/emu/Hokkaido/21B11T/2022	EPI18720830
G2d	18/4/2022	chicken	Akita	Northern	H5N1	A/chicken/Akita/21B1T/2022	EPI18720835
G2d	25/4/2022	emu	Hokkaido	Hokkaido	H5N1	A/emu/Hokkaido/21C1T/2022	EPI18720866
G2d	11/5/2022	emu	Iwate	Northern	H5N1	A/emu/Iwate/21B1T/2022	EPI18720867

## Data Availability

The data that support this study are available from the corresponding author upon reasonable request.

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
