# Peer review of "Genetics of H5N1 and H5N8 High-Pathogenicity Avian Influenza Viruses Isolated in Japan in Winter 2021–2022"

_viruses, 2024, doi:10.3390/v16030358_

Round 1

Reviewer 1 Report

Comments and Suggestions for Authors

This article describes situation on H5 highly pathogenic avian influenza viruses in Japan in winter 2021-2022 and represents their complete phylogenetic analysis, that reveled a strong relationship between viruses from Japan and Siberia, suggesting that the H5 HPAIVs arose from reassortment events with AIVs in Siberia. Migratory wild birds play important role in spread and introduction of HPAIVs to poultry farms, were H5 viruses cause severe outbreaks.  

The study is carried out at high level. The results are accurately described and clearly illustrated in the article.

I have only one remark. There is missed reference number 38 in the line 342.

Reviewer 2 Report

Comments and Suggestions for Authors

The manuscript by Mine and co-authors describe the genetic analysis of HPAI H5N1 viruses isolated during the winter of 2021-2022. The authors found evidence of multiple introductions with viruses of at least three distinct genotypes. One major and several minor issues were identified.

Main concern issue. In addition to the classification of the isolates to the genotype G2a, G2b and G2d which relate specifically to the HA gene, and a brief description in Section 3.3, there is little information on the range of reassortment and possible sources of the other gene segments. This is a shame especially when the authors haave aligned between 10k to over 100k sequences for some segments. Please provide a more detailed description of the reassortment diversity and possibly how they overlay in time and space.

Minor issues

L25. Please insert “first” before the word “detected”.

L40. Please add “in Japan” after the word “occurred”.

L64. The complete name of this commercial software is “CLC Genomics Workbench”

L125. Please replace “occurred” with “detected”.

L179. The sentence here is missing something. It does not read right.

L192. As this is speculation, please insert the word “may” after “Japan”.

L194. The authors do not have any evidence that the exposure was “long term”, please replace with “previous”. Also please insert “more” in front of “prevalence”.

Reviewer 3 Report

Comments and Suggestions for Authors

The manuscript "Genetics of H5N1 and H5N8 high-pathogenicity avian influ- enza viruses isolated in Japan in winter 2021–22" by J Mine at al. is well written, scientifically sound and easy to understand. The implications are important and should be of interest for the readership of "Viruses".

Some minor suggestions:

lines 64-65: Why two different software were used? Are there differences in the outcomes of the analyses with the two software? If yes, please indicates when each software was used.

line 70: Please indicate how the wild bird samples were obtained and selected? Were they processed as described in 2.1 for the poultry farm samples?

line 88: consider replacing "group" with "clade".

line 153: consider moving Figure S1 to main manuscript.
